# The Enhancement of Special-Use Real Estate Properties: The Case of Hospital Facilities

Marta Dell'Ovo [1], Francesca Torrieri [2,3,*], Alessandra Oppio [1], Stefano Capolongo [4], Marco Gola [4] and Andrea Brambilla [4]

[1] Department of Architecture and Urban Studies (DASTU), Politecnico di Milano, Via Bonardi 3, 20133 Milan, Italy; marta.dellovo@polimi.it (M.D.); alessandra.oppio@polimi.it (A.O.)

[2] Department of Architecture, Built Environment and Construction Engineering (DABC), Via Ponzio 31, 20133 Milan, Italy

[3] Department of Industrial Engineering, Università degli Studi di Napoli Federico II, 80138 Naples, Italy

[4] Design & Health Lab, Department of Architecture, Built Environment and Construction Engineering (DABC), Via Ponzio 31, 20133 Milan, Italy; stefano.capolongo@polimi.it (S.C.); marco.gola@polimi.it (M.G.); andrea1.brambilla@polimi.it (A.B.)

* Correspondence: frtorrie@unina.it; Tel.: +39-333-358-0343

**Abstract:** In the Italian context, public investments for the redevelopment and securing of the National Health Service's real estate assets are a crucial topic in the context of the National Recovery and Resilience Plan (NRRP) within the Next Generation Italian strategy. The paper proposes the evaluation of alternative scenarios for accessing financing under the NRRP with respect to the criterion of the technically efficient solution, i.e., the solution that minimizes investment costs while respecting time obligations. The methodology proposed refers to the Cost approach with specific reference to the Depreciated Replacement Cost Method (DRC) in order to estimate the market value in different scenarios. The approach is applied to a case study located in the Piedmont Region, where alternatives are compared with respect to both budget constraints and the timeframe for accessing financing. Given the growing concern for urban regeneration and "public city" rearrangement as an answer to the ongoing global changes, making investments in special-use real estate properties has become a central and challenging issue both in the public and private decision domains.

**Keywords:** hospital; healthcare infrastructures; depreciated replacement cost method; investment decision; enhancement; special use real estate

## 1. Healthcare Assets and Their Challenges in the Italian Context

Hospitals and healthcare facilities are complex infrastructures that embed several functions and multiple users. The hospital's physical setting represents the first step in achieving the desired outcome of high-quality and cost-effective care [1,2].

Given its continuously evolving nature, the health sector requires consistent investments in new and updated facilities and equipment. The Italian scenario highlights the challenges of the high obsolescence rate in healthcare assets, which are estimated to need to be urgently renovated with significant investments [3]. Such contributions offer policymakers scope to shape hospital performance through strategic and financial decisions, although the precise opportunities depend on the ownership, funding, and regulatory systems within the specific facility [4,5]. The cost of healthcare buildings is consistent, and the demand for healthcare services is constantly growing [6]. Therefore, hospitals need to keep functioning as efficient production facilities while, at the same time, incorporating the new need for warm, patient-centered care [7]. The investments required in building or renovating a hospital are usually so significant that the organization must base that investment, along with its programming and design decisions, on a rigorous analysis of the long-term consequences [8]. Healthcare is always seen by developers and investors

as a valuable alternative asset class, with clinics, nursing homes, and assisted homecare reported among the European and Italian top trends in real estate investments; therefore, a high level of quality is requested [9,10]. Especially in the Italian context, the National Recovery and Resilience Plan (Piano Nazionale Ripresa e Resilienza—NRRP) is financing urgent renovation and development of new low-complexity primary care hospital facilities, also known as Territorial Healthcare Centers, Community Health Centers, and Community Hospitals [11–13]. This extraordinary availability of resources enables local administrators, hospital managers, and healthcare decision-makers to look at their assets' renovation strategically, thus increasing the need to evaluate multiple alternatives to intervention. The NRRP has challenging objectives according to the European Commission's purposes: (i) to have a concrete application that is consistent with the defined investment areas and capable of achieving the expected benefits; (ii) to provide implementation schedules aligned with the milestones and deadlines envisaged; (iii) to plan costs consistently with the Plan's budgets (Guidelines for the Implementation of the NRRP 2021 [14]). Furthermore, most European countries are experiencing a progressive aging population. Italy is the first in terms of the increasing proportion of people over the age of 65 that could lead to a higher incidence of chronic degenerative diseases, a greater demand for health and social care, a higher risk of hospital admission, and age-related diseases. At the same time, a staff shortage in acute care hospitals is posing concerns to local and national health policymakers. One possible solution that institutions are approaching to face these challenges is to boost territorial and primary care, along with home care and telemedicine, with the aforementioned typologies of low-care facilities [15,16]. For these reasons, evaluating alternative regeneration scenarios is essential for increasing the transparency of the entire process and meeting the current health demand by highlighting the costs and timing of the investments during the project definition and approval stages, as well as by guaranteeing reimbursement by the European Union.

Moreover, it deserves to be underlined the role of healthcare facilities as promoters of urban regeneration and with a positive impact on different scales of the city. In the Italian panorama, this objective has been elicited by the Italian Ministry of Health through the Ministerial Decree 12/12/2000, better known as the "Veronesi-Piano Project", which established ten parameters that every new hospital should consider for its design. Within the concept of "urbanity", it is very well explained how hospitals should be urban, and their location is not only important from the point of view of accessibility but also to regenerate the urban context. Their presence becomes strategic to achieve specific, sustainable goals. The trend has been emphasized by the document "Hospitals of the Future. A technical brief on re-thinking the architecture of hospitals" was developed by the World Health Organization (WHO) [17] together with the Design and Health Lab of the Polytechnic of Milan with the aim of providing technical guidelines to improve the overall quality of the hospitals' design. In fact, whether located in the city center or at the city boundaries, their "location represent an opportunity for fostering urban regeneration processes" [17,18]. They can be highly integrated in the city context and provide multiple services when located in a central area or work as a connector among several municipalities when located on the periphery.

Given these premises, the article proposes the Depreciated Replacement Cost (DRC) methodology to appraise the market value of a special-use building with the aim of supporting the decision to allocate public funds for community hospital upgrading and safety interventions according to the criterion of technical efficiency, i.e., minimum cost and time. In the absence of a market demand for public real estate, namely for community hospitals, the focus is on the cost and timing of the investment when defining the interventions [19], in order to ensure access to public financing and compliance with the constraints set by the European Union about public expenditure.

The article is therefore structured as follows: Section 2 presents an analysis of the literature on how to appraise the market value of public buildings and evaluate public investments for the construction or redevelopment of hospital facilities in order to highlight

the methodologies most frequently adopted. Section 3 describes the case study; Section 4 illustrates the potential enhancement scenarios; Section 5 discusses the preliminary results; and Section 6 draws some preliminary conclusions.

## 2. The Evaluation of Special Use Real Estate Properties: Methodological Approach and Critical Consideration

### 2.1. Scientific Literature Review

The appraisal of the market value of public buildings, defined by Hajnal and Hajdu [20] as a "quasi-market" segment, i.e., functions with no market demand or highly specific functions, is still challenging, given the lack of comparable or income-based parameters [19,21]. When the asset to be appraised is a healthcare facility, the selection of a suitable valuation method could be more complex due to the lack of standard economic data. Given the criticalities related to the availability of information about the healthcare segment and the choice of the most appropriate appraisal method, a literature review has been developed in the Scopus database using the keywords "market value" AND "public building*" OR "public property*" OR "public asset*". Fifteen papers resulted from the analysis, and after a first screening based on the title and a second one based on the abstract, four contributions have been further investigated and judged to be coherent with the current research objective. Table 1 presents the results of the analysis by giving evidence of the year in which the research has been developed, the country, the typology of building evaluated, the aim to be pursued, and the approach applied.

**Table 1.** Literature review results.

| Authors | Year | Country | Public Building | Aim | Approach Applied |
|---|---|---|---|---|---|
| Tajani and Morano [22] | 2014 | Italy | Religious buildings | To support decisions of Public Administrations involved in the identification of the best decision for the enhancement of public properties | Highest and best use |
| Hajnal and Hajdu [20] | 2017 | Hungary | Museum | To facilitate the market value appraisal of "quasi-market" properties based on uniform principles | Combination of Income approach and Willingness To Pay |
| Battisti and Campo [21] | 2020 | Italy | Landmark | To identify how to apply the Cost approach to the market value appraisal of properties with special features that fall into the "extraordinary" category | Cost approach |
| Fattinanzi et al. [23] | 2020 | Italy | Landmark | Study of real estate appraisal approaches and application of the Depreciated Replacement Cost (DRC) method for valuation of public properties | Cost approach |

The contribution proposed by Tajani and Morano [22] describes how to support negotiations for enhancing public properties in the context of bilateral monopolies and find the market equilibrium between the maximum price the demand is willing to pay and the minimum price the supply is willing to accept. Hajnal and Hajdu [20] focused their research on comparing the pros and cons of combining different methodologies for the appraisal of the market value of a museum by suggesting integrating the Contingent Evaluation (CE) with the determination of the Total Net Revenue. The last two papers proposed by Battisti and Campo [21] and Fattinanzi et al. [23] investigate the potential of

applying the DRC method when the asset to be estimated consists of a public property with special features and no comparable in the market. The papers demonstrate that the DCR approach is recommended, especially when it is not possible to identify income parameters. The DRC is also suggested by the Royal Institution of Chartered Surveyors (RICS) valuation standards. Despite the fact that the number of papers analyzed seems to be inconsistent, these last two contributions are configured as the most coherent to the objective the present paper planned to achieve, and the approach applied is also in line with the property's specificity and the peculiarities of the healthcare sector.

*2.2. The Basis of Value*

The international and European valuation standards [24–26] and the UK Green Book [27] propose different approaches to be used in order to estimate the value of real estate properties in accordance with the basis of value. They are all based on the economic principles of price equilibrium, anticipation of benefit, and substitution.

The principal valuation approaches are:

1.  the Market approach;
2.  the Income approach;
3.  the Cost approach.

According to the International Valuation Standards [24], all these approaches can be used to formulate an opinion of value. In particular, the Market approach can be used where there is an active sales market; the Income approach can be used where there is an active rental market; and the Cost approach can be used where comparable data are unavailable.

The choice of the approach to be used depends on the purpose of the valuation and the specific characteristics of the asset to be valued, as well as on the features of the market.

The valuation of special-use real estate properties with specific reference to the category of hospital facilities opens up stimulating insights from an appraisal point of view concerning the reference market and the methodological approaches used to support investment decisions [28].

In the special case of this paper, where the purpose of the evaluation is to support the choice of a public tenant of purchasing or renovating a property in order to access financing under the NRRP program, there is not a competitive market as a reference, but a peculiar market, such as a monopoly or oligopoly, that makes it challenging to apply the Market approach due to a lack of comparables, or the Income approach based on the revenues obtained from the services provided, due to the public reimbursement mechanism of the Italian healthcare system. Thus, the appraisal approach selected for the case study is the Cost approach.

According to the European Valuation Standards [25] and the Royal Institution of Chartered Surveyors (RICS) Valuation—Global Standards, the principle underlying the Cost approach is that "a purchaser will pay no more for an asset than the cost to obtain one of equal utility, whether by purchase or by construction, unless undue time, inconvenience, risk, or other factors are involved. The approach provides an indication of value by calculating the current replacement or reproduction cost of an asset and making deductions for physical deterioration and all other relevant forms of obsolescence" [26].

Although the use of the Cost approach is debated in the international literature, its relevance and reliability are recognized in specific cases, i.e., in the absence of a competitive market and in the case of special-purpose properties such as hospitals [19]. Among the Cost approaches, the DRC method has been applied to determine the price that a typical buyer would pay, as it is based on replicating the utility of the asset, adjusted for physical, functional, and external deterioration as well as obsolescence.

Thus, the DRC method allows one to estimate the market value of a property by adding the site value to the cost of an equivalent utility asset, minus depreciation, and obsolescence.

Starting from these theoretical considerations, this contribution aims to provide support for evaluating the financial sustainability of transformation or, instead, disposal of a

hospital facility with respect to the public nature of the investment in the context of the NRRP program.

Given the public nature of the investment and the instance of pursuing a balance between economic efficiency and the constraint of minimum cost, the Cost approach with the DRC method has been applied. Special attention has been paid to the depreciation coefficient determination to take into account the age of the asset, its functional obsolescence in response to new community service needs, and the time to realize the intervention.

The proposed methodological approach has been tested on a case study relating to enhancing a healthcare facility in the Piedmont Region, which represents a typical case of a community hospital within the NRRP funding program.

## 3. The Case Study

### 3.1. Description of the Territorial Context and the Healthcare Facility

The hospital is located in a medium-sized city in the Piedmont Region, in a semi-central position near the city's historic center. The city is a town of about 18,000 inhabitants in the Piedmont Region (Figure 1), and it extends over an area of 48.49 km$^2$ with a density of 376.14 inhabits/km$^2$. The city is connected to the province of Alessandria by the 494 Vigevanese state road (SS 494), which connects Milan to Alessandria via Vigevano. It is also served by the railway station located along the Novara-Alessandria railway. The neighborhood where the hospital is located is characterized by mainly residential settlements with public parking, and it is served by the municipal public transport service.

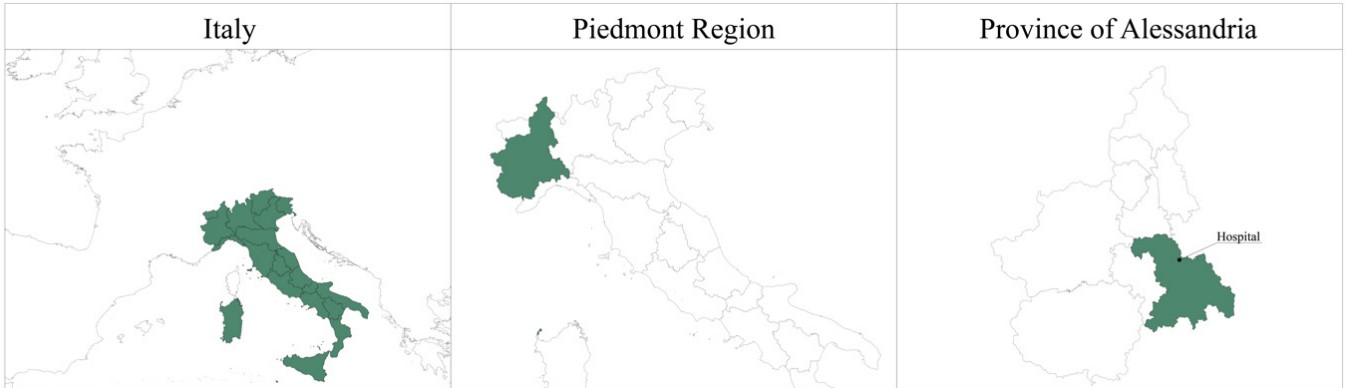

**Figure 1.** Location of the case study. Diagrams designed by the authors.

The property has a healthcare function in which the following services are active: functional recovery and rehabilitation, internal ultrasound and hepatology, analysis laboratories, radiology, and multi-specialist day surgery. It has a Gross Floor Area (GFA) of 9495 sqm, a commercial area of 8283.843 sqm, and a total volume of 31,515 cm.

Figure 2 shows the property's floor plan, while Figure 3 shows some images that represent its current state of maintenance.

As can be seen from the attached figures, the building, whose construction period dates back to 1954, is in a good state of conservation with respect to its structural characteristics, but it has plant equipment and a level of finishes that do not comply with the current performance specifications and regulations of the hospital facilities.

The building consists of seven floors above ground and two underground floors.

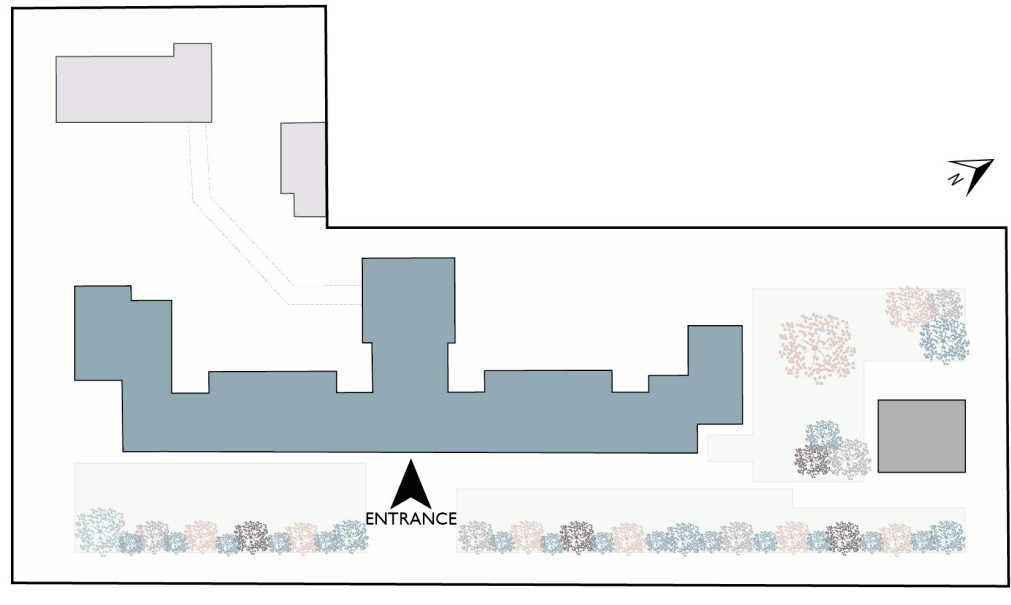

**Figure 2.** Plan of the hospital facility. Diagram designed by the authors.

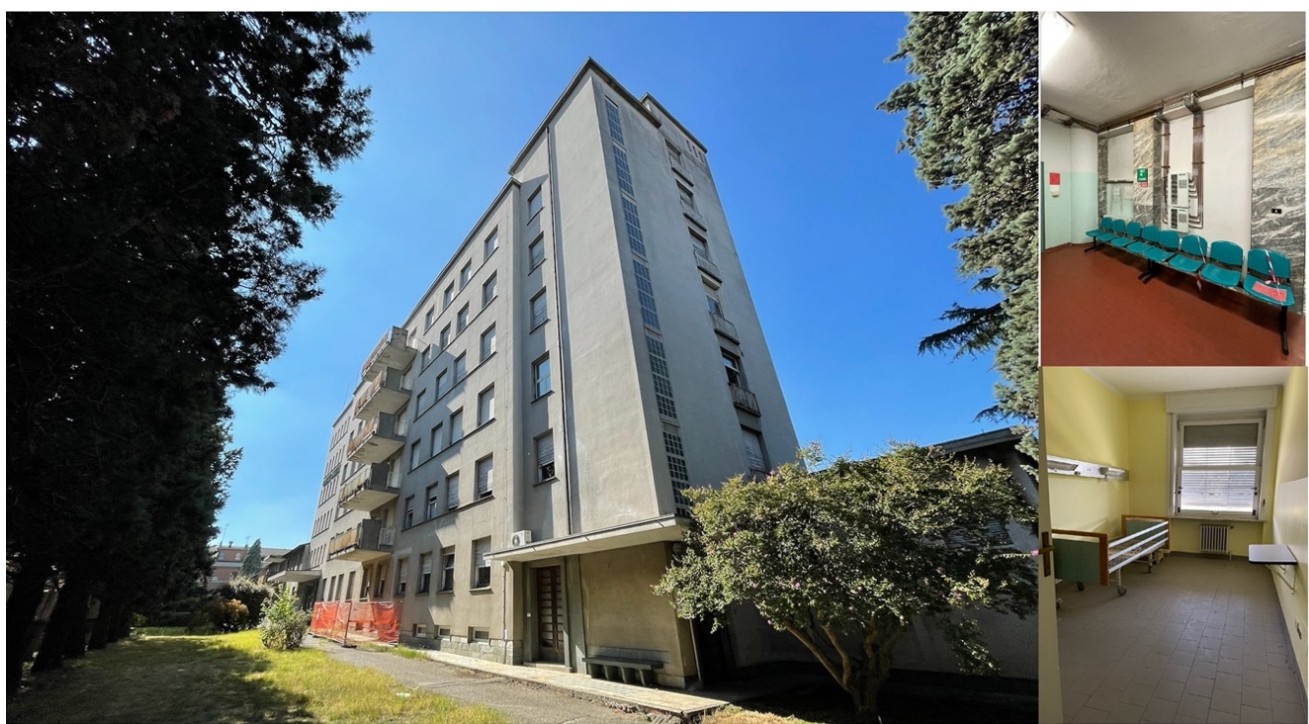

**Figure 3.** Internal and external pictures of the hospital facility. Photos by the authors.

Currently, only five floors out of nine are used: the underground floors, the mezzanine, and the first and second floors. Inside the compendium, a secondary building with a residential function is also used as a caretaker's house.

The proposed case is representative of most of the public Italian health service buildings constructed in the period between 1946 and 1980.

At the national level, the expenditure for maintenance of public hospitals is about 1.4 billion euros per year (80 EUR/m$^2$) for the period 2017–2019, of which 733 million (53%) are ordinary interventions and 660 million (47%) are extraordinary interventions [29].

The decision-making problem therefore concerns the enhancement of the buildings by considering multiple redevelopment alternatives in response to the new community needs, including improving the maintenance conditions of the buildings with respect to the new seismic and energy standards while complying with budget and time constraints imposed by the NRRP.

Under this perspective, different redevelopment scenarios have been analyzed and compared according to the criteria of investment cost and realization time.

*3.2. Enhancement Scenarios*

As explained in Section 3.1, given the asset's characteristics and the fact that it is managed by a public tenant willing to buy the hospital, different scenarios have been generated to compare alternatives by considering as main criteria the investment costs and the time to renovate the building. In detail, in agreement with the regional requirements, the health demands, and the current tenant, a set of possible solutions has been defined considering different levels of complexity: do-nothing (Scenario 0), do-something (Scenario 1), and the intermediary level do-minimum (Scenario 2).

In order to support the investment choice of the tenant, two main scenarios have been identified for the enhancement of the hospital, in addition to maintaining the compendium in its current state.

Scenario 1 is divided into three sub-scenarios—1a, 1b, and 1c—which envisage the preliminary demolition of the health facility and different reconstruction methods.

In particular:

- Scenario 1a consists of the demolition of the entire asset and the reconstruction of a new volume with a gross floor area (GFA) equal to that of the current hospital.
- Scenario 1b, on the other hand, consists of the demolition of the entire asset and the reconstruction of a new volume with GFA limited to the healthcare areas and not currently in operation.
- Scenario 1c envisages the demolition of the entire asset with the consequent reconstruction of a new volume having as GFA one of the healthcare areas and not, as currently in operation, the insertion (or implementation) of new functions and/or healthcare services.

Scenario 2 is divided into two sub-scenarios:

- Scenario 2a, in which the renovation of the existing building includes the cost of seismic safety improvements.
- In Scenario 2b, the existing building is renovated with the insertion (or implementation) of new functions and/or healthcare services in addition to the seismic safety improvement for the entire asset.

Table 2 shows a summary of the scenarios represented in Figure 4.

**Table 2.** Summary of the scenarios.

| | Scenario |
|---|---|
| 0 | Maintenance of the compendium in its current state. |
| 1a | Demolition of the entire asset and reconstruction of a new volume having the same GFA as the current hospital. |

**Table 2.** *Cont.*

| | Scenario |
|---|---|
| 1b | Demolition of the entire asset and reconstruction of a new volume having GFA coinciding with the healthcare areas, and not currently in operation. |
| 1c | Demolition of the entire asset and reconstruction of a new volume having as GFA the one relating to the healthcare areas and not currently in operation, with the insertion (or implementation) of new functions and/or healthcare services. |
| 2a | Seismic retrofit of the asset and renovation work. |
| 2b | Seismic retrofit of the asset and renovation with the insertion (or implementation) of new functions and/or healthcare services. |

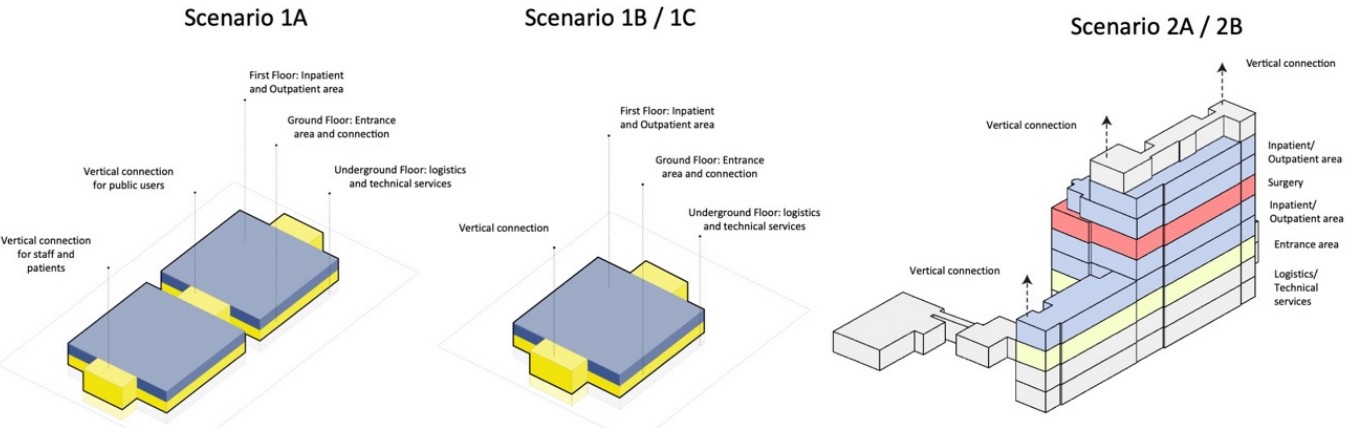

**Figure 4.** Schematization of the different scenarios identified. Diagrams designed by the authors.

*3.3. Methodological Approach*

The Cost Approach

Within this context, on the basis of the special features of a hospital managed by a public tenant, for which there is no direct appreciation from the market and the market is a kind of bilateral monopoly, the market value has been estimated by using the Cost approach.

This procedure is based on the quantification of the total cost that a typical buyer would pay in order to purchase the site and replace an asset with the same functional utility as the existing one, reduced by a coefficient for physical deterioration, for the various types of obsolescence, and for the level of functionality at the time of the estimation.

In fact, the Cost approach is widely applied in professional practice for estimating the market value of old buildings, obsolete sheds, or general economic or business assets for which there is no direct appreciation from the market. The value of the asset can therefore be determined by applying the following formula:

$$\text{Market Value} = V_{site} + [(Ctc + Cic + Fc + Dp) - D] \tag{1}$$

where

- $V_{site}$: Value of the site;
- Ctc: technical construction cost, including suitability works and accommodation;
- Cic: indirect construction costs, including infrastructure costs, concession contributions, professional fees, general, and marketing expenses;
- FC: financial charges on the debt;
- Dp: Developer's profit;
- D: depreciation due to physical deterioration or age; functional and economic obsolescence.

In the appraisal literature, different methods of calculating the depreciation of real estate can be identified, with multiple indications of depreciation coefficients (reduction percentages) to be applied to the value of reconstructing a new building or to the value of industrial machines and plants [30–35].

This depreciation reflects the difference in the typological and technological characteristics, the state of conservation, and the age of the asset being valued compared with a new property taken as comparable.

Regarding the study conducted by Forte and De' Rossi [32], it can be stated that the depreciation due to pure age does not depend only on the current age of the building but also varies with the type of construction. Compared to the age of the building, the two Authors cited logically attribute a more significant gradient of growth in depreciation due to pure age to the buildings built after the middle of the last century, i.e., to those buildings that have experienced the replacement of the traditional masonry load-bearing structure with that, presumably less long-lived, in reinforced concrete.

More recently, this has led to the affirmation [33] that in cases of buildings with complex construction typologies, as in the case under evaluation, given the specific function and the materials used, the analytical calculation of the depreciation must necessarily be carried out by breaking down the building into its "functional and technical elements" (structure, finishes, systems, etc.), since each element is subjected to specific aging and loss of value trends.

According to this approach, the appraisal of the Depreciated Replacement Cost is therefore performed for "functional and technical elements", by assuming that the hospital is a heterogeneous asset, being a whole made up of different functions and technological characteristics and, consequently, being subject to different levels of depreciation.

The estimation of the Depreciated Replacement Cost carried out for the functional and technical elements is divided into the following logically consequential phases:

- Breakdown into the basic functional and technical elements and the cost estimation for replacing the existing asset with a new one with the same utility;
- Definition of the incidence of the cost of the "new" construction of the functional and technical elements—or the homogeneous parts—with respect to the total cost of the asset;
- Definition of a specific depreciation analytical function for each functional and technical component.
- Determination of the Depreciated Replacement Cost for each functional and technical element.
- Aggregation of the depreciated costs of each functional and technical element and determination of the total cost of Depreciated Replacement

The results obtained are described in the following section.

*3.4. Analysis of the Results*

3.4.1. Determination of the Value of the Site

The site where the hospital is located is divided from an urban point of view into two zones with different prominent uses.

The urban planning instruments in force (Municipal Development Plans—MDPs, Municipal Council Resolution No. 87 of 11/13/2013) delimit the area of specific pertinence of the hospital complex as an "Area with infrastructures and equipment for public uses". At the same time, the house and its green area are included in the "Residential area filled and completion areas." Both areas fail within the MDP area called B5.

Given the different urban plan regulations, the following have been considered:

- The determination of the value of the "Residential area, filled and completion zones" at the minimum conventional values of the building areas published for ICI (the municipal property tax) [35] purposes for the homogeneous zone B5. Based on this value and the functional relationship between infrastructural and urban settlement elements, it has been considered that the value of the site with public infrastructure

and services is equal to 30% of the value of the residential area [36]. This reflects the economic principle according to which the utility of an asset depends on the set of characteristics that the asset presents and that, therefore, the value of the areas of the settlement is influenced not only by the relative intrinsic characteristics but also by the extrinsic ones represented precisely by the presence of "Areas with infrastructures and equipment for public use".

- Conversely, the value of the "Residential area filled and completion areas" is considered with reference to the market value of the former caretaker's house, estimated below.

Table 3 below shows the values considered, and the value of the site has been obtained as described in Table 4.

**Table 3.** Value of the site.

| Zones B5 | EUR/sqm |
|---|---|
| Residential area saturated and completion zones | 103 |
| Areas with infrastructure and equipment for public use | 30.9 |

**Table 4.** Estimation value of the site.

| | Description | UM | Input |
|---|---|---|---|
| a | Unit value of the site | EUR/sqm | 30.90 |
| b | Site area | sqm | 9319 |
| c | Covered area former caretaker's house | sqm | 153 |
| d | Net land area (b − c) | sqm | 9166 |
| | **Value of the site (a × d)** | **EUR** | **283,229.40** |

### 3.4.2. Determination of the Construction Cost of the Hospital

The report developed by the IRES research center in the Piedmont Region [37] has been considered to determine the technical construction cost (Table 5). The report ranks healthcare facilities by the level of complexity into three categories: high, medium, and low. The unit cost of the "building box" and the technological plants is between 1800 EUR/sqm and 2500 EUR/sqm.

**Table 5.** Minimum and maximum unit costs for healthcare facilities [37].

| Hospital Complexity | Unit Costs per Square Meter [EUR/sqm] | | |
|---|---|---|---|
| | MINIMUM | REFERENCE | MAXIMUM |
| High | 2000 | 2200 | 2500 |
| Medium | 1900 | 2100 | 2400 |
| Low | 1800 | 2000 | 2300 |

On the basis of the year of construction of the hospital, the current healthcare services, and the characteristics of the buildings, a low level of complexity has been assigned, with a minimum value of 1800 EUR/sqm. This cost has been updated by the Italian National Institute of Statistics (ISTAT, Istituto Nazionale di Statistica) coefficient of 1.088, reflecting the construction cost change from 2017 to 2021 (the year of the analysis).

Therefore, the parametric cost used is equal to 1800 × 1.088 = EUR 1958.40.

According to the assumptions used and the elementary data considered, the technical cost of replacement has been estimated to be equal to:

**Technical cost of replacement = EUR1958.40/sqm × 9485 sqm = EUR18,575,580.67**

Once the technical construction cost (Ctc) was determined, the indirect construction cost (Cic) was also determined.

- Professional fee, considered a percentage of the technical cost of construction with an incidence of 6%;
- operating costs for project management, considered as a percentage of the technical construction cost with an incidence of 2%;
- financial charges estimated as a percentage of 1.30% of the technical construction cost;
- developer's profit, set at 0% of the technical construction cost. The developer's profit (DP) represents the profit that the developer withdraws from the real estate investment. In this case, the developer's profit is considered null, as the owner has the sole institutional purpose of maintaining the value of its real estate portfolio without pursuing any kind of profit.

Table 6 shows the final calculation of the replacement cost.

**Table 6.** Estimation of the replacement cost.

|   | Description | UM | Input |
|---|---|---|---|
| a | Technical cost of new construction | EUR | 18,575,580.67 |
| b | Soft costs (6%) | EUR | 1114,534.84 |
| c | Management charges (2%) | EUR | 371,511.60 |
| d | Financial charges (1.3%) | EUR | EUR241,482.55 |
| e | Profit of the promoter (0%) | EUR | - |
| **k** | **Cost of the replacement (a + b − c + d + e)** | **EUR** | **20,303,109.67** |

As can be seen from Table 6, the cost of replacement is equal to the

**Cost of the replacement = EUR20,303,109.67**

3.4.3. Depreciation Functions Related to the Functional Elements of the Hospital

To specify the depreciation functions corresponding to the functional and technical elements of the hospital under investigation, the total replacement cost has been broken down according to the functional and technical elements in compliance with the UNI 8290 standard—Residential construction, technological system, and terminological classification.

The compendium was analyzed using the following categories as a reference:

- structures;
- finishes;
- installations.

Once the building has been divided into its basic components and the replacement cost and incidence have been determined, a suitable depreciation function has been specified for each of the functional and technical elements identified.

Physical depreciation refers to the loss of value as described by the following function:

$$\Delta C_{d\log} = \left( (C_0 - Vr) \times \frac{(1+i)^n - 1}{(1+i)^v - 1} \right) \tag{2}$$

where

$\Delta C_{d\log}$ = Physical depreciation.
$C_0$ = Initial value.
$Vr$ = Salvage value at the end of the functional and technical components service lives.
$i$ = Discount rate.
$V$ = Number of years of the effective life of the functional and technical components (service life).
$N$ = Number of years in the past life of the functional and technical elements on the date of appraisal (useful life).

The inputs of the depreciation function were taken from literature (Table 7) and detailed with respect to the specific characteristics of the case under examination.

**Table 7.** Parameters used as the basis for the calculation of the depreciation coefficients of the functional and technical elements of a building [33].

| Functional Element | Years | | | |
|---|---|---|---|---|
| | a | b | c | D (%) |
| Masonry structure | 80–120 | 300 | 50 | 25 |
| Structure in reinforced concrete (minimum exposed parts) | 60–65 | 120 | 40 | 30 |
| Structure in reinforced concrete (face view) | 55–60 | 120 | 40 | 40 |
| Floors in reinforced concrete and brick | 80–90 | 120 | 40 | 12 |
| Wooden structures | 60–65 | 120 | 40 | 30 |
| Roofs, insulation, and waterproofing | 25–35 | 90 | 4 | 8 |
| Sheet metal works (gutters, downspouts) | 10 | 25 | 5 | 18 |
| Plasters and false ceilings | 37–40 | 75 | 15 | 18 |
| Floors | 55–60 | 40 | 20 | 10 |
| Paintings | 5 | 5 | - | 100 |
| Windows and iron works | 45–50 | 80 | 15 | 10 |
| External wooden windows | 15 | 40 | 5 | 20 |
| Interior doors | 35–40 | 40 | 12 | 15 |
| Electric-specials | 35 | 35 | - | 100 |
| Elevators | 35–40 | 60 | 5 | 5 |
| Plumbing, heating, and fire fighting | 33 | 40 | 10 | 5 |
| Conditioning | 15 | 15 | - | 100 |
| External accommodations | 35 | 60 | 20 | 25 |

Where

a. useful life of the functional and technical elements without extraordinary maintenance interventions;
b. useful life of the functional and technical elements under the hypothesis that they are subjected to regular maintenance interventions;
c. frequency of maintenance interventions;
d. the cost of periodic extraordinary maintenance is determined as a percentage of the replacement cost of the building.

The parameters determined for the hospital compendium are shown in Table 8.

**Table 8.** Input parameters for depreciation determination (years).

| Works Description | Useful Life | Service Life |
|---|---|---|
| Structures and foundations | 67 | 65 |
| Works in reinforced concrete | 67 | 65 |
| Roof | 67 | 60 |
| Internal and external partitions | 67 | 60 |
| Floors and walls | 67 | 60 |
| Plasters | 67 | 40 |
| Paintings | 67 | 5 |
| External windows | 67 | 15 |
| Internal windows | 67 | 40 |
| Sanitary | 67 | 40 |
| Elevator systems | 67 | 40 |
| Air conditioning systems | 32 | 15 |
| Sanitary water systems | 32 | 33 |
| Electrical and special systems | 22 | 35 |

As can be seen from Table 8, the structural components have been divided according to the materials used (reinforced concrete). The depreciation of the structures has logically been limited to the physical features, thus excluding the functional obsolescence related to the possibility of introducing new products on the market capable of replacing existing materials.

The economic life span of the reinforced concrete structure is equal to its durability. Nonetheless, it appears clear that the service life of such a structure depends on numerous factors, such as good and correct practice in the design and construction phases, as well as periodic inspection and maintenance. On the basis of research and practice, the total service life of a reinforced concrete building subjected to ordinary and extraordinary maintenance interventions can be assumed to be equal to 120 years and to 65 years without a maintenance plan [33].

In this case, the technical survey has revealed the poor conservation of the reinforced concrete structures, whose useful life has been assumed to be 65 years. Furthermore, by considering the age of the hospital, built in 1954, it is also necessary to consider that the reinforced concrete structures should be adapted to the last seismic safety requirements in line with Ministerial Decree No. 58 of 02/28/2017.

Therefore, it has been assumed to have a service life of 65 years and a zero residual value.

The roof structural elements, made of reinforced concrete, have been assimilated into reinforced concrete structures in terms of "durability". Also, in this case, the absence of regular extraordinary maintenance was assumed, and, in consideration of the exposed face, a service life of 60 years was assumed.

The finishing elements (painting, plastering, flooring, etc.) usually have a shorter useful life than the structures; in this case, considering the level of deterioration surveyed, a depreciation equal to the new reconstruction cost was assumed.

Regarding the technological systems, the installation period has been determined with reference to the documents provided by the tenant, maintenance interventions included. Therefore, a useful life equal to that of the existing plants has been assumed.

The discount rate used in the corresponding depreciation function reflects the variation of the construction cost estimated for the specific functional element from 1955 (time of construction) to the year 2022, the year of the appraisal, and, therefore, to an extent equal to 1.26% on average/year, based on the ISTAT revaluation index.

The depreciation for each functional and technical element was estimated on the basis of the analyses carried out.

The results obtained are reported in Table 9.

**Table 9.** Hospital compendium depreciation by functional and technical elements.

| Works Description | Depreciation | % Incidence on Total Depreciation |
|---|---|---|
| Structures and foundations | EUR 3,269,302 | 19% |
| Works in reinforced concrete | EUR 1,727,529 | 10% |
| Roof | EUR 650,145 | 4% |
| Internal and external partitions | EUR 1,225,988 | 7% |
| Floors and walls | EUR 1,318,866 | 8% |
| Plasters | EUR 835,901 | 5% |
| Paintings | EUR 910,203 | 5% |
| External windows | EUR 835,901 | 5% |
| Internal windows | EUR 854,477 | 5% |
| Sanitary | EUR 724,448 | 4% |
| Elevator systems | EUR 650,145 | 4% |
| Air conditioning systems | EUR 1,969,012 | 11% |
| Sanitary water system | EUR 679,922 | 4% |
| Electrical and special systems | EUR 1,671,004 | 10% |

As can be seen from Table 9, the total depreciation of the hospital is equal to the following:

**Total depreciation cost: EUR 17,322,844.34**

Therefore, the market value of the hospital at present, on the basis of the Depreciated Replacement Cost method, is obtained as the sum of the following inputs: value of the site, cost of a building of the same utility, depreciation, technical expenses, financial charges, management costs, and the developer's profit (Table 10).

**Table 10.** Estimate of the market value of the hospital unit.

| | Description | UM | Input |
|---|---|---|---|
| a | Site value | EUR | 283,229.40 |
| b | Total cost of new construction | EUR | 20,303,109.67 |
| c | Depreciation total | EUR | −17,322,844.34 |
| | **Market value of the hospital structure at present (a + b — c )** | **EUR** | **3,263,494.74** |

## 4. Determination of the Investment Value of the Re-Functionalization Scenarios

### 4.1. Scenario 1

The investment value of the different scenarios was estimated with respect to the demolition and reconstruction costs under the various hypotheses as well as the market value of the asset in the current conditions.

The demolition cost, including landfill charges, was estimated according to the work item reported by the Public Works Price List of the Piedmont Region (ed. 2021) [38], work item code 01.A02.A05. 030, as explained in Table 11.

**Table 11.** Estimate of demolition costs.

| | Description | UM | Input |
|---|---|---|---|
| a | Unit cost for the complete demolition of buildings | EUR/cm | 12.84 |
| b | Empty volume for full | cm | 32,808 |
| c | Total cost of demolition (axb) | EUR | 421,254.72 |
| d | Cost of landfill charges | EUR/kg | 0.009 |
| e | Average weight of rubble per unit of PPV | kg/cm | 573.16 |
| f | Total cost of landfill charges | EUR | 169,238.10 |
| | **Total demolition cost (c + f)** | **EUR** | **590,492.82** |

The landfill costs were estimated by considering the average weight of the rubble per unit of empty volume per total kg/cm equal to 573,16 (White Paper on private reconstruction outside the historic centers in the municipalities affected by the Abruzzo earthquake of 6 April 2009) and the fees established by Art. 15, paragraph 1, letter b, of Regional Law 1/2018 of the Piedmont Region for landfill charges for hazardous waste.

Table 11 shows the estimated demolition costs of EUR 590,492.82.

The reconstruction cost of the different Scenarios was estimated with reference to what has already been described in Section 3.4.2, Tables 5 and 6.

The following tables provide a summary of the appraisal stages for each scenario (Tables 12–15).

**Table 12.** Cost of building hospital structure in Scenario 1a.

| | Description | UM | Input |
|---|---|---|---|
| a | Unit cost of construction | EUR/sqm | 1958.40 |
| b | GFA | sqm | 9485.08 |
| c | Technical cost of construction (a × b) | EUR | 18,575,580.67 |
| d | Soft costs (6%) | EUR | 1,114,534.84 |
| e | Operating costs (2%) | EUR | 371,511.6 |
| f | Financial charges (1.3%) | EUR | EUR241,482.55 |
| g | Developer's profit (0%) | EUR | - |
| | **Total construction cost (c + d + e + f + g)** | **EUR** | **20,303,109.67** |

**Table 13.** The construction cost of the Scenario 1b community hospital.

| | Description | UM | Input |
|---|---|---|---|
| a | Unit cost of construction of the Community hospital | EUR/sqm | 1,958.40.80 |
| b | GFA | sqm | 3000 |
| c | Technical cost of construction (a × b) | EUR | 5,875,200.00 |
| d | Soft costs (6%) | EUR | 352,512.00 |
| e | Operating costs (2%) | EUR | 117,504.00 |
| f | Financial charges (1.3%) | EUR | 76,377.60 |
| g | Developer's profit (0%) | EUR | - |
| | **Total construction cost (c + d + e + f + g)** | **EUR** | **6,421,593.60** |

**Table 14.** Community hospital construction cost in Scenario 1c.

|  | Description | UM | Input |
|---|---|---|---|
| a | Unit cost of construction of the Community hospital | EUR/sqm | 2285.80 |
| b | GFA | sqm | 2000 |
| c | Technical cost of construction (a × b) | EUR | 4,569,600.00 |
| d | Soft costs (6%) | EUR | 274,176.00 |
| e | Operating costs (2%) | EUR | 91,392.00 |
| f | Financial charges (1.3%) | EUR | 59,404.80 |
| g | Developer's profit (0%) | EUR | - |
| | **Total construction cost (c + d + e + f + g)** | **EUR** | **4,994,572.80** |

**Table 15.** Cost of building the community house in Scenario 1c.

|  | Description | UM | Input |
|---|---|---|---|
| a | Unit cost of construction Community house | EUR/sqm | 1958.40 |
| b | GFA | sqm | 1000 |
| c | Technical cost of construction (a × b) | EUR | 1,958,400.00 |
| d | Soft costs (6%) | EUR | 117,504.00 |
| e | Operating costs (2%) | EUR | 39,168.00 |
| f | Financial charges (1.3%) | EUR | 25,459.20 |
| g | Developer's profit (0%) | EUR | - |
| | **Total construction cost (c + d + e + f + g)** | **EUR** | **2,140,531.20** |

The extraordinary maintenance cost of the former caretaker's house has been determined by the DEI Building Types price list—Tipografia del Genio Civile, Edition 2019 [39], typology A12—medium and high-end residential construction, renovation of a residential building in the central area, with a cost of EUR 1357.07/sqm.

Therefore, the cost of renovating the former caretaker's house is presented in Table 16.

**Table 16.** Cost of renovation of the former caretaker's house.

|  | Description | UM | Input |
|---|---|---|---|
| a | Unit cost of renovation | EUR/sqm | 1357.07 |
| b | GFA | sqm | 306 |
| | **Total construction cost (a × b)** | **EUR** | **415,262.87** |

The estimated total value of the investment for Scenarios 1a, 1b, and 1c is shown in Tables 17–19 below.

**Table 17.** Total investment value in Scenario 1a.

|  | Description | UM | Input |
|---|---|---|---|
| a | Market value of the Territorial Presidium | EUR | 3,495,825.7 |
| b | Demolition costs | EUR | 590,492.82 |
| c | Cost of rebuilding to new | EUR | 20,303,109.67 |
| d | Cost of renovation of the former caretaker's house | EUR | 415,262.87 |
| | **Total investment value (a + b + c + d)** | **EUR** | **24,804,691.13** |

**Table 18.** Total investment value in Scenario 1b.

|   | Description | UM | Input |
|---|---|---|---|
| a | Market value of the Territorial Presidium | EUR | 3,495,825.7 |
| b | Demolition costs | EUR | 590,492.82 |
| c | Cost of rebuilding to new | EUR | 6,421,593.60 |
| d | Cost of renovation of the former caretaker's house | EUR | 415,262.87 |
| | **Total investment value (a + b + c + d)** | **EUR** | **10,923,175.06** |

**Table 19.** Total investment value in Scenario 1c.

|   | Description | UM | Input |
|---|---|---|---|
| a | Market value of the Territorial Presidium | EUR | 3,495,825.7 |
| b | Demolition costs | EUR | 590,492.82 |
| c | Cost of building the community hospital | EUR | 4,994,572.80 |
| d | Cost of building the community house | EUR | 2,140,531.20 |
| e | Cost of renovation of the former caretaker's house | EUR | 415,262.87 |
| | **Total investment value (a + b + c + d + e)** | **EUR** | **11,636,685.46** |

*4.2. Scenario 2*

Scenario 2 is divided into two sub-Scenarios, 2a and 2b, which provide seismic upgrading of the entire building and different renovation interventions.

In particular, the value of the investment in Scenario 2a is given by the current market value of the asset, to which are added the costs of the seismic upgrading of the hospital, the geotechnical tests, the lighting renovation, and the renovation costs of the former caretaker's house.

The value of the investment in Scenario 2b is instead given by the market value of the hospital, to which are added the costs of the seismic upgrading, the geotechnical tests, the hard renovation intervention, as well as the costs of renovating the former caretaker's house.

The determination of the individual cost items that contribute to the investment cost of Scenario 2 is shown below.

4.2.1. The Cost of Seismic Retrofitting

The determination of the seismic safety improvement cost has been based on an analysis of the gray literature concerning the parametric costs of interventions to make buildings safe. In particular, reference has been made to the contribution proposed by Cosenza et al. [40] and the Ministerial Decree No. 58 28/02/2017 Annex A: Guidelines for the classification of seismic risk of constructions (Allegato A: Linee guida per la classificazione del rischio sismico delle costruzioni) http://www.mit.gov.it/normativa/decreto-ministeriale-numero-58-del-28022017 (20 November 2021)

According to the indications of the Italian guidelines, the seismic adaptation of a reinforced concrete building depends on the seismic risk class of the building in its current state compared with that envisaged by the legislation. The seismic risk class is a function of two parameters: one for structural safety, called the Safety Index, and an economic parameter, called the Expected Average Annual Loss. In the absence of information on the current risk class of the property being valued, a prudential adjustment was assumed to improve the safety index by two or more classes.

Concerning this type of intervention, the parametric cost reported by the literature [40] is equal to 343 EUR/sqm.

The estimated seismic upgrading costs are shown in Table 20.

**Table 20.** Seismic retrofit cost.

| | Description | UM | Input |
|---|---|---|---|
| a | Unit cost of seismic retrofitting | EUR/sqm | 370.78 |
| b | GFA | sqm | 9485.08 |
| | **Total cost of seismic retrofitting (a × b)** | **EUR** | **3,516,906.42** |

4.2.2. The Cost of Geotechnical Investigations

The expenses for the geotechnical and structural investigations were estimated by considering the "White Paper on private reconstruction outside the historic centers in the municipalities affected by the Abruzzo earthquake of 6 April 2009" ("Libro bianco sulla ricostruzione privata fuori dai centri storici nei comuni colpiti dal sisma dell'Abruzzo del 6 Aprile 2009") and were calculated at 12 EUR/sqm.

The costs of the estimated geotechnical investigations are shown in Table 21.

**Table 21.** Cost of geotechnical investigation.

| | Description | UM | Input |
|---|---|---|---|
| a | Unit cost of geotechnical tests | EUR/sqm | 14.16 |
| b | GFA | sqm | 9485.08 |
| | **Total construction cost (a × b)** | **EUR** | **134,308.73** |

4.2.3. Determination of the Cost of Renovation of Hospital Facilities

For the determining the cost of renovating hospitals, a basic reference is represented by the report "The implementation of the extraordinary program for building renovation and technological modernization of healthcare assets" ("L'attuazione del programma straordinario per la ristrutturazione edilizia e l'ammodernamento tecnologico del patrimonio sanitario") [41] developed by the Court of Auditors (Corte dei Conti), Central Control Section on the management of public administrations, which divides the renovation costs of hospitals into "hard, medium and soft" depending on the type of intervention as it is described by the following table, which shows the parametric costs based on data developed and processed by the Ministry of Health (Table 22).

**Table 22.** Reference costs for new buildings and renovations.

| Hospitals | Reference cost (EUR/sqm) |
|---|---|
| New construction/expansion | +2200.00 |
| Hard renovation | 1850.00 |
| Medium renovation | 1300.00 |
| Soft renovation | 800.00 |

The soft renovation costs (Scenario 2a) and the hard renovation costs (Scenario 2b) were estimated with respect to the data reported in Tables 23 and 24.

**Table 23.** Hospital soft renovation cost.

| | Description | UM | Input |
|---|---|---|---|
| a | Hospital soft renovation unit cost | EUR/sqm | 1357.07 |
| b | GFA | sqm | 9485.08 |
| | **Total construction cost (a × b)** | **EUR** | **8,202,697.18** |

**Table 24.** Hospital hard renovation costs.

| | Description | UM | Input |
|---|---|---|---|
| a | Hospital hard renovation unit cost | EUR/sqm | 1999.85 |
| b | GFA | sqm | 9485.08 |
| | **Total renovation cost (a × b)** | **EUR** | **18,968,737.24** |

The total value of the investment estimated for Scenarios 2a and 2b is shown in Tables 25 and 26.

**Table 25.** Total investment value in Scenario 2a.

| | Description | UM | Input |
|---|---|---|---|
| a | Market value of the Territorial Presidium | EUR | 3,495,825.7 |
| b | Seismic retrofit costs | EUR | 3,516,906.42 |
| c | Cost of geotechnical investigations | EUR | 134,308.73 |
| d | Soft renovation cost | EUR | 8,202,697.18 |
| e | Cost of renovation of the former caretaker's house | EUR | 415,262.87 |
| | **Total investment value (a + b + c + d + e)** | **EUR** | **15,756,000.97** |

**Table 26.** Total investment value in Scenario 2b.

| | Description | UM | Input |
|---|---|---|---|
| a | Market value of the Territorial Presidium | EUR | 3,495,825.7 |
| b | Seismic retrofit costs | EUR | 3,516,906.42 |
| c | Cost of geotechnical investigations | EUR | 134,308.73 |
| d | Hard renovation costs | EUR | 18,968,737.24 |
| e | Cost of renovation of the former caretaker's house | EUR | 415,262.87 |
| | **Total investment value (a + b + c + d + e)** | **EUR** | **26,531,041.02** |

## 5. Discussion of the Results

Based on the scenarios identified and described in Section 3, the investment value has been estimated by considering the purchase at the current market value of the hospital as invariant. To synthesize the analysis developed below, it is possible to appreciate the steps performed for each scenario.

- Scenario 0: the main phase elaborated considers the appraisal of (1) the market value of the existing hospital;
- Scenario 1: as already mentioned, it is further divided into three sub-scenarios involving the preliminary demolition of the healthcare facility and the reconstruction;
- Scenario 1a: the main phases consider the appraisal of (1) the market value of the existing hospital; (2) the demolition cost of the existing hospital; (3) the construction of a new hospital by considering the actual GFA (~9500.00 sqm); (4) the renovation cost of the former caretaker's house;
- Scenario 1b: the main phases elaborated consider the appraisal of (1) the market value of the existing hospital; (2) the demolition cost of the existing hospital; (3) the construction of a community hospital, reducing the actual GFA (3000.00 sqm); (4) the renovation cost of the former caretaker's house;
- Scenario 1c: the main phases elaborated consider the appraisal of (1) the market value of the existing hospital; (2) the demolition cost of the existing hospital; (3) the construction of a community hospital decreasing the actual GFA (2000.00 sqm); (4) the construction of a community house (1000.00 sqm); (5) the renovation cost of the former caretaker's house.

Scenario 2 is further divided into two sub-scenarios involving the seismic retrofitting of the healthcare facility and renovation interventions:

- Scenario 2a: the main phases consider the appraisal of (1) the market value of the existing hospital; (2) the seismic retrofit cost; (3) the cost of geotechnical tests; (4) the soft renovation costs; (5) the renovation cost of the former caretaker's house;
- Scenario 2b: the main phases elaborated consider the appraisal of (1) the market value of the existing hospital; (2) the seismic retrofit cost; (3) the cost of geotechnical tests; (4) the hard renovation costs; and (5) the renovation cost of the former caretaker's house.

Table 27 shows the investment values of the scenarios considered.

**Table 27.** Total investment value.

| Total Investment Value (EUR) | | | | |
|---|---|---|---|---|
| Scen_ 1a | Scen_ 1b | Scen_ 1c | Scen_ 2a | Scen_ 2b |
| 24,804,691.13 | 10,923,175.06 | 11,636,685.46 | 15,756,000.97 | 26,531,041.02 |

Together with the total investment costs, the times needed to implement each scenario have been estimated based on a schedule of the planned interventions shown in Table 28.

**Table 28.** Total investment value and time schedule.

| | Total Investment Value (EUR) | Times |
|---|---|---|
| SCENARIO 0 | 3,495,825.76 | 6 months |
| SCENARIO 1a | 24,804,691.13 | 54 months (5 years, 6 months) |
| SCENARIO 1b | 10,923,175.06 | 45 months (3 years, 9 months) |
| SCENARIO 1c | 11,636,685.46 | 45 months (3 years, 9 months) |
| SCENARIO 2a | 15,756,000.97 | 52 months (5 years, 2 months) |
| SCENARIO 2b | 26,531,041.02 | 56 months (5 years, 8 months) |

From the comparison between the total investment costs and the implementation time for each scenario, it can be seen that apart from Scenario 0, which consists of leaving the structure in its current conditions, therefore underused and not compliant with seismic and performance regulations, Scenario 1b, i.e., the demolition of the entire asset and the reconstruction of a new volume with the same size as the current one, has a lower investment cost but requires the same time as Scenario 1c.

For this reason, the demolition and reconstruction of the building are more convenient than an intervention to adapt and renovate the existing structure.

All the previously described scenarios involve the temporary replacement of the current functions to allow construction work. Given the potential buyer's intention to not interfere with the daily healthcare services during the construction phases, in both Scenarios 1, demolition and reconstruction, and Scenarios 2, seismic retrofitting and renovation, the temporary rental of additional spaces has not been considered in the overall estimation, nor have the costs for moving operations since they are a common variable among all the different options. In addition, the possibility of temporarily occupying public buildings free of charge has been made explicit.

This conclusion seems to align with the most recent research, which highlights that the state of conservation in Italian hospitals is in a critical condition. It is estimated that more than two-thirds of the hospital assets are in the final stages of their life cycle, and over half do not comply with the new functional and operational instances.

## 6. Conclusions and Future Development

The proposed methodology allows one to estimate the market value of a public healthcare facility on the basis of the Cost approach—the DRC method.

In order to support the decision of the public tenant to purchase and transform the building to make it more functional in compliance with the current standards, the needs of the community, and the opportunity to access the NRRP program, different scenarios have been analyzed with respect to the investment costs and construction times as the main decision variables.

The results obtained show, in line with the most recent research, that the state of conservation and the type of hospital suggest are the drivers of the strategy to be adopted, i.e., the re-functionalization of the existing asset or rather the demolition and reconstruction according to more resilient and flexible schemes, able to respond to the current paradigm of the hospital of the future [4,42].

The DRC method, based on functional elements, allows for an estimate of the market value of the building in the absence of a specific market reference, such as in many public contexts and special-use buildings. The analysis is a first tentative attempt to compare multiple scenarios of intervention by applying methodologies based on the cost criterion.

Moreover, the study of the current state of conservation of the building and the evaluation of the potential enhancement scenarios have supported the decision of the tenant to buy the building in order to have access to the NRRP program and to respond to the "performance" achievement imposed by the founding program in terms of costs and time respect.

On the other hand, recent studies [43] prepared for the European Investment Bank have quantified the investment gap in the sector at about 32 billion, thus being much higher than the financial resources derived from the NRRP for seismic upgrades, health technologies, community homes, and hospitals. About 58% of the National Health Service's property stock was built before 1970, and therefore, it is one of the most energy-intensive assets at the community level and poorly aligned with the ongoing evolution of the service models.

One of the potential responses to this additional demand for financial resources may be the involvement of private entities through forms of public-private partnerships (PPP).

PPP can find application, as far as real estate is concerned, either for the construction of Community Homes and Hospitals, for the implementation of energy efficiency interventions, for the construction/renovation/refurbishment of basic or level I/II hospitals, or for the construction of poles to host services to be dedicated to health basins.

In addition to the assessment of the investment costs and the timing of the interventions, the NRRP requires qualitative analysis of the environmental and social impacts of the investment in order to account for the added value of the investment for the community. This aspect concerning the evaluation of community benefits can be considered a further line of research to develop the study beyond an economic and financial perspective.

**Author Contributions:** Conceptualization, S.C., A.O. and F.T.; methodology, A.O. and F.T.; validation, A.O. and F.T.; formal analysis, M.D.; investigation, A.B., M.D. and M.G.; resources, A.B., M.D. and M.G.; data curation, A.B., M.D. and M.G.; writing—original draft preparation, A.B., M.D., M.G., A.O. and F.T.; writing—review and editing, A.B., M.D., A.O. and F.T.; visualization, A.B., M.D. and M.G.; supervision, S.C., A.O. and F.T.; project administration, S.C. All authors have read and agreed to the published version of the manuscript.

**Funding:** This research received no external funding.

**Data Availability Statement:** All the useful data analyzed and developed by authors are reported in the paper. The authors refer to all the useful reports and publications where the readers can find additional data related to other healthcare facilities with different level of complexity (medium and high care). For this reason, any other documents are not linked to this research.

**Conflicts of Interest:** The authors declare no conflict of interest.

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
