# Peer review of "The Enhancement of Special-Use Real Estate Properties: The Case of Hospital Facilities"

_land, doi:10.3390/land12081638_

Round 1

Reviewer 1 Report

This piece of case study is important from two aspects: first, it is a good overview of the current stage of Italian public sector investment; second, renewal of the non-finished scientific debate about the valuation of public properties. 

Only concern Authors may consider: analysis scenarios are not supported with a general overview of development potentials. Readers may interest about the process how final scenarios were filtered.

"Valorization" is not a common phrase within valuation literature, Authors may use another term instead of it. 

It is a well-written and useful article, suggested to publish.

Author Response

This piece of case study is important from two aspects: first, it is a good overview of the current stage of Italian public sector investment; second, renewal of the non-finished scientific debate about the valuation of public properties. 

Thank you very much.

Only concern Authors may consider: analysis scenarios are not supported with a general overview of development potentials. Readers may interest about the process how final scenarios were filtered.

According to your comment the scenario generation has been better explained in section 3.2.

"Valorization" is not a common phrase within valuation literature, Authors may use another term instead of it. 

Thanks for the suggestion, “valorization” has been changed in “enhancement”.

It is a well-written and useful article, suggested to publish.

Thank you again.

Reviewer 2 Report

The reviewed article successfully presents the evaluation of alternative scenarios for accessing to financing under the NRRP with respect to the criterion of the technically efficient solution. The presented assessment techniques are exhaustive and allow for a comprehensive assessment of various variants of modernization of a public utility building that performs functions in the field of health care.
Nevertheless, the presented text lacks (at least a short) discussion of one more important aspect of the described problem. Well, a long-term analysis of the effectiveness of any investments, including and perhaps especially public ones, requires an assessment of the future demand for services generated by the modernized infrastructure. In the case of health care services, this demand is largely dependent on the course of demographic processes. Most EU countries, including Italy, are experiencing a progressive aging problem. This in turn, as is known, will result in a constant increase in demand for health and social care services.
Therefore, when planning the scale and thus evaluating the costs of investment in health care infrastructure, this issue should also be taken into account. Otherwise, the considered investment variants may turn out to be insufficient (and perhaps sometimes rescaled) in relation to future needs, which in the long term will require further investments, thus increasing the total costs and lowering the overall economic efficiency.
It is therefore worth, at least at the level of theoretical considerations, to supplement the presented analyzes with indications of methods and techniques to estimate and take into account the future demand for public services in the case of investments financed from central or local government budgets.

None

Author Response

The reviewed article successfully presents the evaluation of alternative scenarios for accessing to financing under the NRRP with respect to the criterion of the technically efficient solution. The presented assessment techniques are exhaustive and allow for a comprehensive assessment of various variants of modernization of a public utility building that performs functions in the field of health care.

Thank you for your comment.

Nevertheless, the presented text lacks (at least a short) discussion of one more important aspect of the described problem. Well, a long-term analysis of the effectiveness of any investments, including and perhaps especially public ones, requires an assessment of the future demand for services generated by the modernized infrastructure. In the case of health care services, this demand is largely dependent on the course of demographic processes. Most EU countries, including Italy, are experiencing a progressive aging problem. This in turn, as is known, will result in a constant increase in demand for health and social care services.

Therefore, when planning the scale and thus evaluating the costs of investment in health care infrastructure, this issue should also be taken into account. Otherwise, the considered investment variants may turn out to be insufficient (and perhaps sometimes rescaled) in relation to future needs, which in the long term will require further investments, thus increasing the total costs and lowering the overall economic efficiency.
It is therefore worth, at least at the level of theoretical considerations, to supplement the presented analyzes with indications of methods and techniques to estimate and take into account the future demand for public services in the case of investments financed from central or local government budgets.

Thank you very much for your suggestions. Considering your comment, we know that this is a strategic challenge that is being addressed with these newer types of more territorial healthcare to address the needs of an increasingly elderly population and that this is considered in the theoretical framework of our scenarios. We add a comment in the introduction.